# Spatiotemporal Relationships between Ecosystem Health and Urbanization on the Tibetan Plateau from a Coupling Coordination Perspective

**Yu Hu [1], Tong Wu [2,*], Luo Guo [1,*] and Shidong Zhang [1]**

1   College of Life and Environmental Sciences, Minzu University of China, Beijing 100081, China; 21302358@muc.edu.cn (Y.H.); 21400281@muc.edu.cn (S.Z.)
2   The Natural Capital Project, Stanford University, Stanford, CA 94301, USA
*   Correspondence: tongwu@stanford.edu (T.W.); guoluo@muc.edu.cn (L.G.); Fax: +86-10-68931632 (L.G.)

**Abstract:** A complex relationship exists between natural environmental and urban systems. This study focuses on the province of Qinghai, which is a major part of the Tibetan Plateau. For the period 1995–2020, we explore the relationships between ecosystem health and urbanization. We established an indicator system for ecosystem health and urbanization and quantitatively analyzed the spatiotemporal pattern of coupling coordination between the two. It shows that between 1995 and 2020, the urbanization level in Qinghai showed a continuous upward trend, with the urbanization index in 2020 increasing by 147.20% compared to 1995. The level of ecosystem health also showed an upward trend, with the ecosystem health index (EHI) value in 2020 increasing by 3.31% compared to 1995. The coupling coordination degree between ecosystem health and urbanization in Qinghai increased year-on-year. Areas with high coupling coordination between ecosystem health and urbanization basically overlapped with areas with high urbanization, and the area gradually increased during the study period. Locations with slightly unbalanced development were located on the periphery of high coupling-coordination areas, and that area expanded in the years covered in this study. The low coupling-coordination areas were located in the northwest and southwest of Qinghai, bounded by the Qinghai-Tibet Highway. These results can support the evaluation of ecosystem health and urbanization on the Tibetan Plateau and high-altitude river source areas similar to Qinghai in China and elsewhere, contributing to sustainable land use policy.

**Keywords:** ecosystem health; urbanization; land use change; Pressure–State–Response model; Tibetan Plateau



## 1. Introduction

Ecosystems provide humans with various service functions in terms of both natural resources and the living environment, and ecosystem service functions are the foundation of human survival and development. An ecosystem can only provide long-term services to human society if it maintains the integrity of its structure and functions and has the ability to resist interference and recover [1]. Ecosystem health refers to the ability of an ecosystem to maintain the integrity of its structure and function after being affected by human factors, as well as its ability to continuously meet the reasonable needs of human society [2,3]. This is an intuitive understanding of the quality of a given ecosystem [4]. Protecting the integrity and health of ecosystems is the foundation of ecological protection work. Ecosystems provide the basic conditions for human survival; people can directly or indirectly benefit from these conditions [5]. Therefore, a comprehensive assessment of ecosystem service functions and ecosystem health can inform sustainable development [6]. However, as the breadth and intensity of human activities continue to expand, global ecological conditions are changing at an unprecedented pace and scale. In particular, China's urbanization has expanded significantly in recent decades [7]. Between 1995 and 2020, the urbanization

rate of permanent residents had risen from 29.04% to 63.89% [8], accompanied by various land use patterns [9]. In the context of accelerated urbanization, the overexploitation of land will cause severe eco-environmental problems such as water loss and soil erosion, biodiversity loss, and frequent natural disasters. This leads to significant changes in the structure and function of ecosystems, exacerbating their degradation [10–13], posing a significant threat to the survival and development of human society. When the condition of the ecosystem deteriorates, improving its health is generally considered the goal of ecological management. Balancing socio-economic development and ecosystem protection has become an urgent issue in certain regions of China. How urbanization can develop within the carrying capacity of the eco-environment, and how the eco-environment can provide a space for urbanization, have become crucial issues [14,15].

Ecosystem health is considered one of the most important issues in ecosystem management, playing a core role in comprehensive ecosystem assessment [16]. It enables us to understand the state of ecosystems and the threshold of ecological degradation from both the structure and function of ecosystems [3]. Different ecosystem health assessment methods can provide effective tools for evaluating regional ecological environment conditions. Assessing ecosystem health is to diagnose its state and explore its driving factors [17], in order to identify areas with prominent ecological problems, ensure ecological security, improve resource management levels, and provide a strong foundation for ecosystem governance and restoration. Ecosystem health assessments have been conducted at different scales, involving multiple levels, including global [18], regional [19], landscape [20], and ecosystem levels [21]. For example, Ran et al. conducted an assessment of the health status of global ecosystems and explored factors affecting regional differences in ecosystem health [18]; Meng et al. assessed the health status of rural ecosystems in Jiangsu Province, China [22]; Kang et al. evaluated the ecosystem health status of urban agglomerations [23]; Cheng et al., Liu et al., and Meng et al. assessed the ecological health of river ecosystems, wetland ecosystems, and forest ecosystems at the regional scale [24–26].

Currently, there are three main types of assessment frameworks for ecosystem health, namely, subsystem evaluation, the VORS model (V represents vigor, O represents organization, R represents resilience, and S represents ecosystem service) [18,27], and the PSR model (P is pressure, S is state, R is response) [28,29]. Since it is equally important to assess the integrity and health of ecosystems, this study uses the PSR model constructed by the Organization for Economic Co-operation and Development (OECD) and the United Nations Environment Program (UNEP) to integrate the assessment of ecosystem services into the assessment system of ecosystem health, with a view to comprehensively assess the health and sustainable development of Qinghai's ecosystem [30]. The central point of the PSR model is that the ecological degradation will lead to qualitative and quantitative changes. Society should respond to these changes through economic and environmental protection [30]. The PSR model provides a clear causal relationship, which can then be used to examine environmental issues from a human-centered perspective and has therefore been widely used in ecological assessments [29].

The Tibetan Plateau is a vast area with an average altitude of more than 4000 m, and it is known as the "third pole of the world" [31]. Qinghai Province occupies the northeast of the Plateau [32] and is the headwaters of the Yangtze, Yellow, and Lantsang Rivers [33]. It is therefore often referred to as the "Water Tower of China" [34]. Qinghai boosts a diverse range of ecosystem types [31] and is both ecologically and economically important to its region and to the country [35–37]. Nevertheless, the eco-environment in Qinghai is fragile [38], and its geographical position, environmental conditions, population distribution, and other factors affect the process of regional urbanization. The urban development level in Qinghai remains relatively low, but exhibits high volatility. Moreover, economic and social development is highly concentrated in the three major central cities of Xining, Geermu, and Delingha [39]. However, in recent years, China has vigorously developed its western region, resulting in a significant increase in population migration to Qinghai and significant economic and social development. As economic growth continues and

urbanization intensifies, the issue of unreasonable development of the natural environment is becoming increasingly prominent in Qinghai [38]. Quantitative analysis of the coupling coordination development between ecosystem health and urbanization could therefore be helpful in improving Qinghai's land management. Few existing ecosystem health assessment frameworks have combined ecosystem integrity with human demand for ecosystem services. This study establishes an indicator system for urbanization as well as ecosystem health, evaluates the ecosystem health status using the PSR model, and quantitatively analyzes the spatiotemporal pattern of the coupling coordination development between ecosystem health and urbanization in Qinghai during 1995–2020.

## 2. Materials and Methods

### 2.1. Study Area

Qinghai Province (31°39′–39°19′ N, 89°35′–103°04′ E) is located in northwest inland China and covers an area of $7.2 \times 10^5$ km$^2$ [40]. Qinghai has towering mountains, diverse terrain [41], crisscrossing rivers, and scattered lakes (Figure 1). Qinghai lies on the Tibetan Plateau [42], the highest altitude in the world [43]. Qinghai is a province with a large number of ethnic minorities. As many as 43 ethnic minorities live here. The ethnic minorities that live for generations are mainly Tibetan, Hui, Tu, Salar, and Mongols [44]. In recent years, driven by the western development strategy, the construction of the "the Belt and Road", and the rural revitalization strategy, Qinghai's economy has been growing and its population has been increasing [45]. The proportion of urban population increased by 26.2% from 1995 to 2020, and urbanization is in the stage of accelerated development [8].

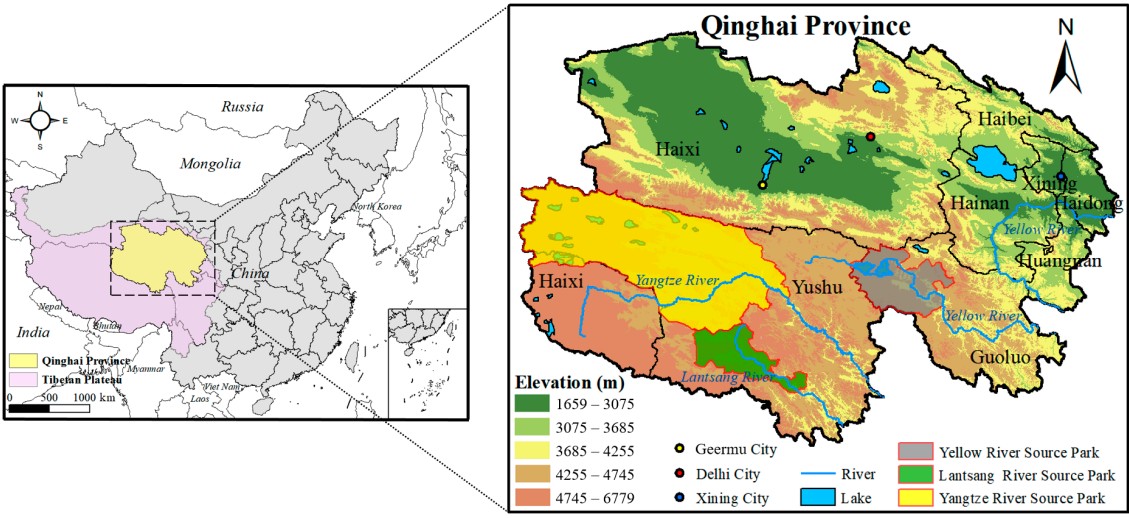

**Figure 1.** Location of the study area.

### 2.2. Data Collection

Remote sensing data used to quantify ecosystem health level include: China's national scale multi-period land use/land cover thematic data with a resolution of 1 km, which is obtained from the Resource and Environment Science Data Center (RESDC) of the Chinese Academy of Sciences, using Landsat satellite remote sensing images as the main information source and through manual visual interpretation (https://www.resdc.cn/, accessed on 1 January 2023); resolution of 1 km of China's population spatial distribution kilometer grid dataset, obtained from RESDC; and on the basis of SPOT/VEGETATION PROBA-V 1KM PRODUCTS vegetation index data, an annual normalized differential vegetation index (NDVI) spatial distribution dataset with a resolution of 1 km was obtained through concatenation, embedding, and projection transformation, which was obtained from RESDC. The remote sensing dataset used to quantify urbanization level is the Prolonged Artificial Nighttime-Light Dataset of China (PANDA) with a resolution of 1 km, obtained

from the Tibetan Plateau Data Center (TPDC) (https://data.tpdc.ac.cn/zh-hans/data/e755f1ba-9cd1-4e43-98ca-cd081b5a0b3e, accessed on 1 January 2023). This data applies the Nighttime-Light convolution Short-long Term Memory (NTLSTM) network to grow the world's first set of Prolonged Artificial Nighttime-Light Dataset of China (PANDA). The model evaluation of the model and the original image shows that the average Root Mean Square Error (RMSE) reaches 0.73, the R-Square ($R^2$) reaches 0.95, and the linear slope of the pixel level is 0.99. The data quality of the generated product is high.

*2.3. Methods*

2.3.1. Methods of Assessing LULC Changes

A land use transfer matrix and LULC change direction model (*LCDM*) are used to evaluate land use change in Qinghai. Land use transfer matrix models can visualize the spatial and temporal changes of LULC [46]. This can be achieved in ArcGIS 10.4 (Esri, Redlands, CA, USA).

*LCDM* can evaluate the impact of LULC change on ecosystem function [47], which can provide a basis for ecosystem health assessment. The ecological levels weight of different land use types are shown in Table 1. This is achieved through the *LCDM*, which uses the formula:

$$LCDM = \frac{\sum_{i=1}^{n} \left[ A_{ij} \times (D_j - D_i) \right]}{A} \times 100\% \tag{1}$$

where *LCDM* refers to land use change value, $\in [1, n]$. $A_{ij}$ refer to the area of land use change, $D_i$ and $D_j$ refer to the ecological value of land use types before and after, respectively. A reflects the total transformed area. *LCDM* > 0 means that land use transformation is developing in a favorable direction, while *LCDM* < 0 is the opposite.

**Table 1.** Ecological levels weight of different land use types.

| LULC | Farmland | Forest | Grassland | Water Body | Built-Up Land | Unused Land |
|---|---|---|---|---|---|---|
| Ecological levels | 0.105 | 0.246 | 0.146 | 0.344 | 0.082 | 0.077 |

2.3.2. Assessing Ecosystem Health Based on the PSR Model

In this study, the PSR model evaluated the ecosystem health of Qinghai in 1995, 2000, 2005, 2010, 2015, and 2020. Based on the PSR model [48], we constructed an evaluation system of EHI by using land use, NDVI, and population data. The pressure dimension of PSR model includes two indicators: population density (PD) and landscape fragmentation index (LFI). And the state dimension includes five indicators: normalized difference vegetation index (NDVI), landscape diversity index (LDI), average patch area index (APAI), ecosystem service value (*ESV*) and ecological resilience (*ER*) (Table 2). The Analytic Hierarchy Process (AHP) method and Monte Carlo method were used to identify the weight of the index [49].

The product of the value of each index and its weight is the EHI. When evaluating the multi-index system, it is necessary to standardize the original index data to ensure the feasibility of the results. The formula is as follows:

$$HI = \sum_{i=1}^{n} w_i x_i \tag{2}$$

where *i* indicates the number of indicators, $x_i$ indicates the value of a specific indicator, and $w_i$ indicates the weight of the *i*-th indicator.

2.3.3. Indicators of the PSR Framework

Both indicators PD and NDVI were obtained from the Resource and Environment Science Data Center (RESDC) of the Chinese Academy of Sciences. The indicators LFI, LDI, and APAI used to evaluate EHI were obtained by Fragstats 4.2 software.

**Table 2.** The Pressure–State–Response (PSR) framework and the weight of each indicator.

| Target Level | Standard Level | Weight | Indicator Level | Weight |
|---|---|---|---|---|
| Ecosystem health index (EHI) | Pressure | 0.3 | Population density (PD) | 0.20 |
| | | | Landscape fragmentation index (LFI) | 0.10 |
| | State | 0.7 | Normalized difference vegetation index (NDVI) | 0.24 |
| | | | Landscape diversity index (LDI) | 0.09 |
| | | | Average patch area index (APAI) | 0.05 |
| | | | Ecosystem service value (*ESV*) | 0.22 |
| | | | Ecological resilience (*ER*) | 0.10 |
| | Response | --- | Restored areas | ---- |

This paper chose the approach based on unit value [50] to analyze *ESV*. We divided ecosystems into six categories based on the international land use and cover change (LUCC) classification standards and the actual conditions of the study area: farmland, forest, grassland, water body, build-up land, and unused land. Then, the expert-based method for ecosystem services valuation developed by Xie et al. [51] was adopted, which utilizes the similar classification from Costanza et al. [52] and is most widely used in China. The unit *ESV* is 1/7 of the economic value of food production service from farmland [53]. *ESV* is calculated as follows:

$$ESV_{s,k,t} = \alpha_{k,t} \times VC_{s,k} \tag{3}$$

where $\alpha_{k,t}$ refers to the area of land use type $k$ at time $t$. $VC_{s,k}$ indicates the per-unit value coefficient of ES, and $k$ is land use type; $t \in \{1995, 2000, 2005, 2010, 2015, 2020\}$.

This paper uses *ER* to quantify the ability of the ecosystem to maintain its own functions and its state unchanged when it was disturbed by the outside world using the following equation:

$$ER = D \sum_{i=1}^{n} S_i \times E_i \tag{4}$$

where $D$ represents the landscape diversity index; $S_i$ is the area ratio of each landscape type; $E_i$ is the resilience value of the landscape type $i$ [54].

2.3.4. Evaluation Urbanization Index

Luminous remote sensing can reflect intensity of human activity, including social economy and energy consumption. Zhang et al. proposed a nighttime-light convolution long-term and short-term memory network and applied it to derive a prolonged artificial nighttime-light dataset of China (PANDA) [55]. In view of the extreme correlation between PANDA and socio-economic indicators, this study uses PANDA to refer to the level of urbanization in Qinghai.

2.3.5. The Coupling Coordination Degree Model

Coupling coordination degree (CCD) refers to degree of interaction and influence between multiple systems or elements [13]. The formula used to calculate CCD between ecosystem health and urbanization is as follows:

$$C = \left\{ \frac{f(\alpha) \times f(\beta)}{\left[ \frac{f(\alpha) + f(\beta)}{2} \right]^2} \right\}^{\frac{1}{2}} \tag{5}$$

$$T = af(\alpha) + bf(\beta) \tag{6}$$

$$D = \sqrt{C \times T} \tag{7}$$

where $C$ is the system coupling degree, $f(\alpha)$ is urbanization level, $f(\beta)$ is ecosystem health level, $T$ is the total system score of urbanization level and ecosystem health level, reflecting the overall synergy effect of urbanization and ecosystem health. $n$ is the adjustment coefficient ($n \geq 2$), and this paper sets $n$ as 2. Given the equal importance of urbanization and ecosystem health, $a = b = 0.5$. $C \in [0, 1]$.

The coupling coordination model can reflect the coordinated development level of urbanization and ecosystem health. $D$ is the coupling co-scheduling of the system, $D \in [0, 1]$. As the value of $D$ is larger, the CCD between subsystems is higher, and vice versa [56]. In view of Qinghai's high ecological vulnerability and low carrying capacity, this study adopted urbanization classification standards that are consistent with the actual situation in Qinghai (Table 3).

**Table 3.** CCD level and coordinated characteristic between urbanization level and ecosystem health level.

| CCD Range | CCD Level | Subsystem Characteristic | Coordinated Characteristic |
|---|---|---|---|
| $0.1 \geq D > 0$ | Seriously unbalanced development | $f(\alpha) - f(\beta) > 0.1$ <br> $f(\beta) - f(\alpha) > 0.1$ <br> $\lvert f(\alpha) - f(\beta) \rvert \leq 0.1$ | Ecosystem health lagged <br> Urbanization lagged <br> Synchronously development |
| $0.2 \geq D > 0.1$ | unbalanced development | $f(\alpha) - f(\beta) > 0.1$ <br> $f(\beta) - f(\alpha) > 0.1$ <br> $\lvert f(\alpha) - f(\beta) \rvert \leq 0.1$ | Ecosystem health lagged <br> Urbanization lagged <br> Synchronously development |
| $0.3 \geq D > 0.2$ | Slightly unbalanced development | $f(\alpha) - f(\beta) > 0.1$ <br> $f(\beta) - f(\alpha) > 0.1$ <br> $\lvert f(\alpha) - f(\beta) \rvert \leq 0.1$ | Ecosystem health lagged <br> Urbanization lagged <br> Synchronously development |
| $0.4 \geq D > 0.3$ | Moderately balanced development | $f(\alpha) - f(\beta) > 0.1$ <br> $f(\beta) - f(\alpha) > 0.1$ <br> $\lvert f(\alpha) - f(\beta) \rvert \leq 0.1$ | Ecosystem health lagged <br> Urbanization lagged <br> Synchronously development |
| $D > 0.4$ | Highly balanced development | $f(\alpha) - f(\beta) > 0.1$ <br> $f(\beta) - f(\alpha) > 0.1$ <br> $\lvert f(\alpha) - f(\beta) \rvert \leq 0.1$ | Ecosystem health lagged <br> Urbanization lagged <br> Synchronously development |

## 3. Results

### 3.1. Changes in Temporal and Spatial Heterogeneity in LULC

The evolution of land use in Qinghai is revealed by the land use transfer matrix (Figure 2). During 1995–2020, land use in Qinghai changed significantly. We selected grassland and forest to represent the level of the ecosystem health, as well as farmland and build-up land to represent the level of urbanization, for analysis, and use the dotted circles to display areas with significant changes. From 1995 to 2020, among converted land use types, grassland was the largest, reaching 97,140 km$^2$, accounting for about half of the total land use transfer area, and this transformation occurred throughout Qinghai. The area converted from other land use types to forest accounts for 8.16% of the total land use, and the converted areas are in the east and south of Qinghai. The area converted from other LULC types to farmland and build-up land is relatively small, and the areas transferred out are concentrated in the urban areas of Qinghai. The *LCDM* can characterize the change of land use, which is helpful in further exploring the impact on ecological functions. The results were as follows: $LCDM_{1995-2000} = 0.15\%$, $LCDM_{2000-2005} = 5.26\%$, $LCDM_{2005-2010} = 9.04\%$, $LCDM_{2010-2015} = 5.76\%$, $LCDM_{2015-2020} = 1.40\%$. During the period from 1995 to 2020, the *LCDM* value in Qinghai first increased and then decreased.

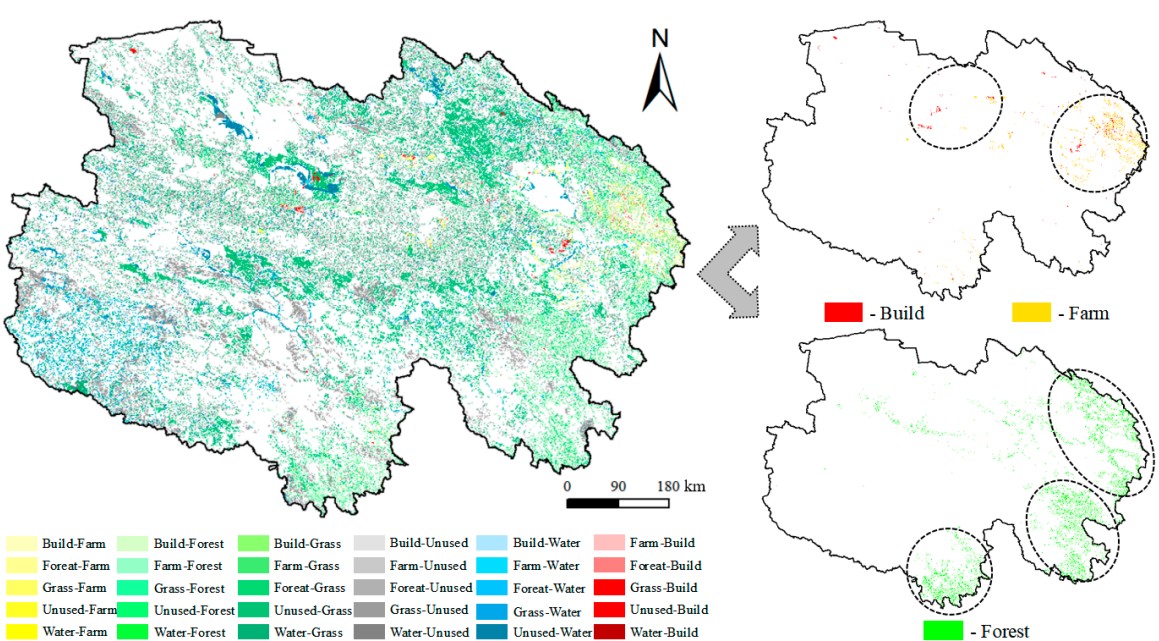

**Figure 2.** Transfer matrix map of land use in Qinghai from 1995 to 2020.

*3.2. Evaluating Ecosystem Health*

3.2.1. Changes in the Pressure and State Indicators

Overall, from 1995 to 2020, the value of pressure level indicators (PD and LFI) in Qinghai showed a downward trend, indicating that pressure levels are increasing (Table 4). By analyzing the trend of changes in LFI values separately, it can be concluded that the degree of landscape fragmentation in the study area sharply increased and then eased. Figure 3 shows that during the study period, the difference between the population density in eastern Qinghai and the surrounding areas gradually narrowed, but the area with the largest population density is still concentrated in Xining and its surrounding areas. The degree of landscape fragmentation in Qinghai Lake, Sanjiangyuan National Park, and northern Qinghai is relatively low, while in areas with frequent human activities such as eastern Qinghai, the degree of landscape fragmentation is relatively high.

**Table 4.** Changes in different indices from 1995 to 2020 in Qinghai.

| Indicator Level | Index Level | Year | | | | | |
|---|---|---|---|---|---|---|---|
| | | 1995 | 2000 | 2005 | 2010 | 2015 | 2020 |
| Pressure | PD | 0.0024 | 0.0019 | 0.0026 | 0.0011 | 0.0007 | 0.0009 |
| | LFI | 0.0128 | 0.0106 | 0.0106 | 0.0107 | 0.0110 | 0.0113 |
| | Total | 0.0152 | 0.0125 | 0.0132 | 0.0118 | 0.0117 | 0.0122 |
| State | NDVI | 0.0668 | 0.0618 | 0.0678 | 0.0665 | 0.0707 | 0.0714 |
| | LDI | 0.0156 | 0.0149 | 0.0149 | 0.0149 | 0.0149 | 0.0160 |
| | APAI | 0.0108 | 0.0090 | 0.0090 | 0.0090 | 0.0090 | 0.0111 |
| | *ESV* | 0.0345 | 0.0351 | 0.0354 | 0.0355 | 0.0358 | 0.0371 |
| | *ER* | 0.0173 | 0.0166 | 0.0165 | 0.0165 | 0.0166 | 0.0177 |
| | Total | 0.1450 | 0.1374 | 0.1436 | 0.1424 | 0.1470 | 0.1533 |
| EHI | - | 0.1602 | 0.1499 | 0.1568 | 0.1542 | 0.1587 | 0.1655 |

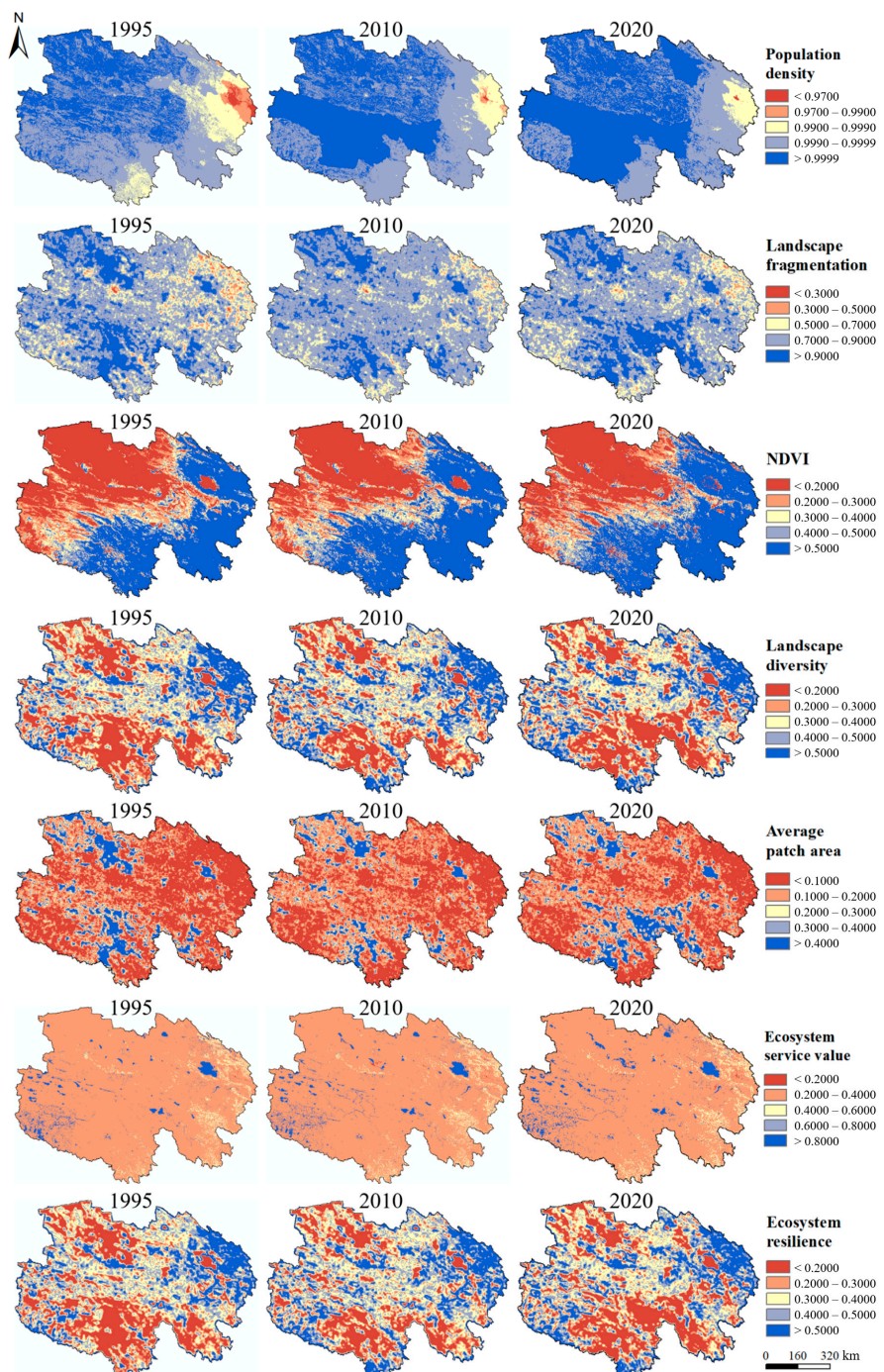

**Figure 3.** The spatial distribution of different indices for the years 1995, 2010, and 2020.

The state level indicators (NDVI, LDI, APAI, *ESV*, and *ER*) in Qinghai fluctuated and rose during the study period and reached the peak in 2020. Qinghai reached its best ecosystem health state in 2020 (Table 4). The spatial distribution of NDVI and *ESV* is similar, with high value areas in the eastern and southern regions and low value areas in the rest of the province (Figure 3). The ER value is similar to the LDI value in spatial distribution. High values are concentrated in northeast and southwest of Qinghai and around Qinghai Lake; a low value is distributed in the south and northwest of Qinghai and Qinghai Lake, while the distribution trend of APAI is exactly opposite to the spatial distribution of ER and LDI.

3.2.2. Changes in the Ecosystem Health Index

The EHI value showed a downward trend during 1995–2000 and 2005–2010, with a large decline of 6.43% in 1995–2000 and a slight decline of 1.66% in 2005–2010. In the rest of the study period, the EHI values showed a trend of growth, and the growth rate was the largest between 2000 and 2005, reaching 4.60%, followed by 2015–2020, with the growth rate at 4.28%. The growth rate from 2010 to 2015 was relatively small at 2.92% (Table 4).

In order to facilitate observation and insight into the spatial distribution of EHI during the study, EHI is divided into seven levels (Figure 4). The overall distribution trend of EHI is that the east and south of Qinghai are high-value areas, the northwest is low value, and the rest are transitional areas. With the passage of time, EHI value in the east of Qinghai gradually increased, especially in Qinghai Lake, Zhaling Lake, Eling Lake, and other lakes and surrounding areas. Analyzing the area ratio of EHI at all levels can show the changes of EHI in Qinghai more comprehensively. Among the I-VII levels, the proportion of level IV is the largest, followed by levels V, III, VI, and VII, and the proportion of level I and II is very small. Levels III, IV, and V can represent moderate ecosystem health status. During 1995–2000, the area of levels III, IV, and V increased slightly, but decreased after 2000. On the contrary, the area of level VI and VII, which represent good ecosystem health, decreased from 1995 to 2000 but increased after 2000, indicating that the ecosystem health of Qinghai is improving.

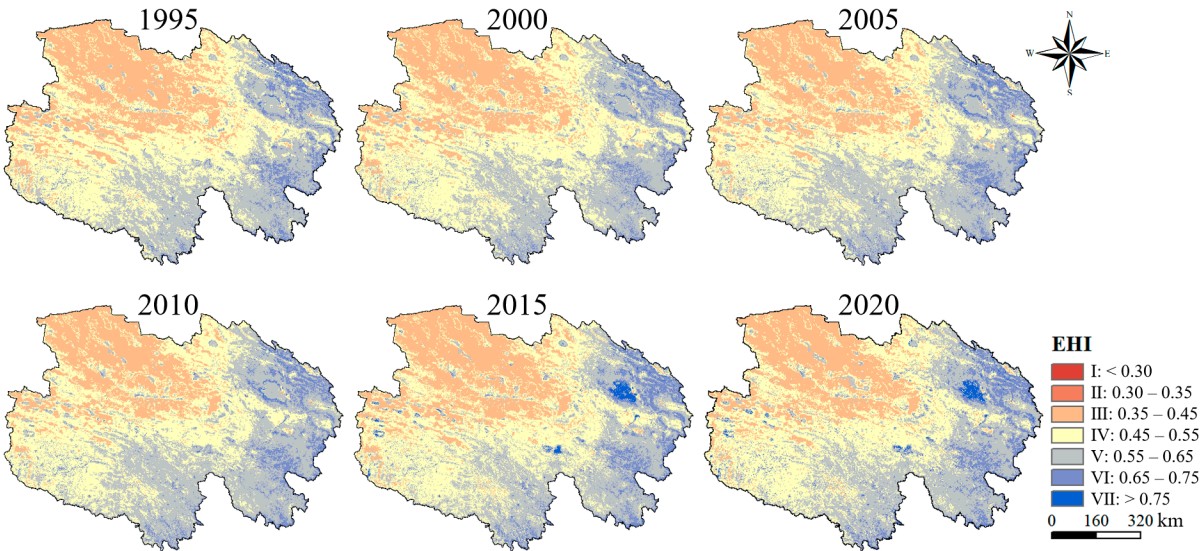

**Figure 4.** Assessment of the EHI for Qinghai.

*3.3. Changes in Urbanization Index*

During 1995–2020, urbanization level in Qinghai gradually improved. Figure 5 shows the regional distribution of urbanization in Qinghai. Proportion of low- and lower-level urbanization areas reached over 95%. Qinghai's urbanization level is generally not high. From 1995 to 2020, the area of low- and lower-level urbanization areas in Qinghai decreased by 3.63%; the area of medium-level urbanization areas significantly increased by 450.41%; the area of high- and higher-level urbanization areas showed a consistently upward trend, with a total increase of 441.61%. In 2020, the area of high- and higher-level urbanization areas in Qinghai reached its maximum, and the regional urban system gradually improved. With the continuous acceleration of urbanization development in Qinghai, high-level regions are radiating and expanding outward, the process centered around the three major cities of Xining, Geermu, and Delingha. It is worth mentioning that during the research period, the urbanization level of small and medium-sized cities and some counties in Qinghai also significantly increased. Overall, the dynamic changes of urbanization level in

Qinghai are influenced by transportation and terrain. The level of urbanization is negatively correlated with altitude and positively correlated with transportation level.

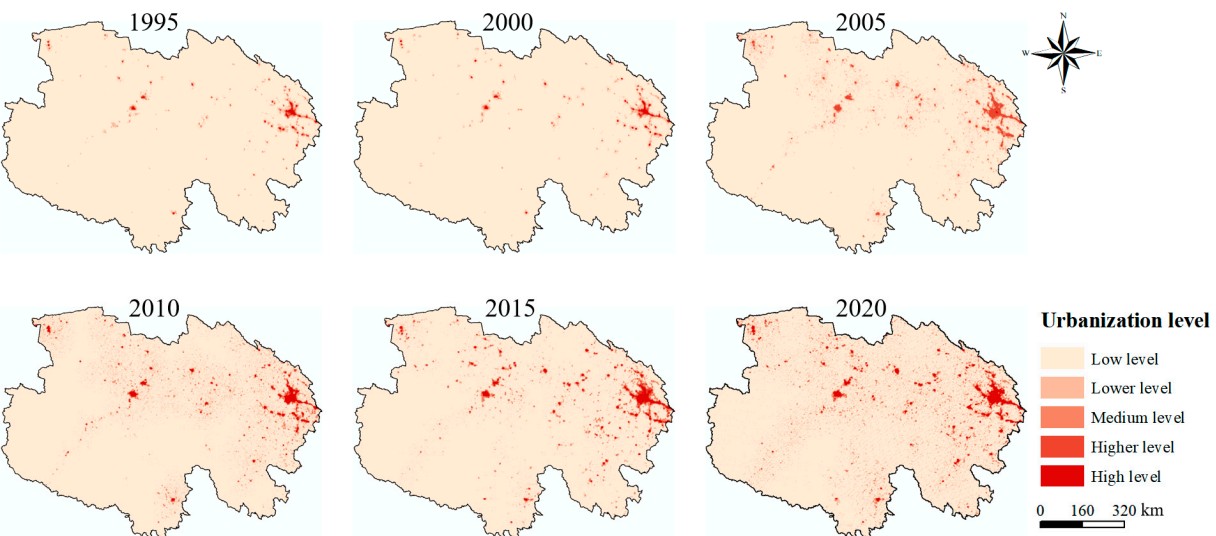

**Figure 5.** Spatial distribution of urbanization level in Qinghai from 1995 to 2020.

### 3.4. Coupling Coordination between Ecosystem Health and Urbanization

As shown in Figure 6a, from 1995 to 2020, the *D* and *C* value of urbanization and ecosystem health in Qinghai showed an upward trend. Although the *D* value shows a continuous upward trend, the increase is small. The growth rate was positive during 2000–2005 and 2010–2015, with the largest growth rate of 23.93% during 2000–2005. The growth rate decreased between 2005–2010 and 2015–2020, with the smallest growth rate of 5.30% between 2005–2010. Compared with the *D* value, the *C* value increased significantly, showing a positive growth in 2000–2005 and 2015–2020, with the largest growth rate of 29.02% in 2000–2005. The growth rate decreased between 2005–2010 and 2010–2015, with the smallest growth rate of 5.53% between 1995–2000.

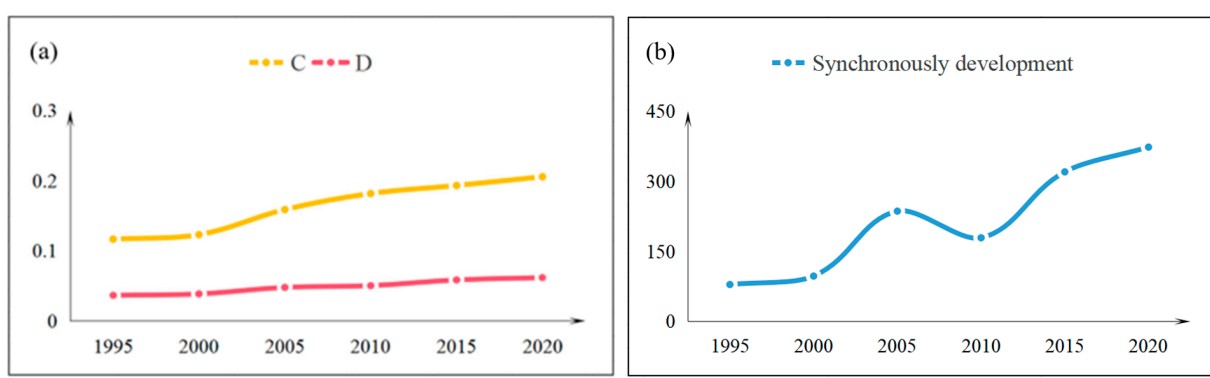

**Figure 6.** (**a**) Evolution of the *C* and *D* values between urbanization and ecosystem health in Qinghai during 1995–2020. (**b**) Evolution of the grid area of coordination between urbanization and ecosystem health.

The coordinated development of ecosystem health and urbanization in Qinghai changed from 1995 to 2020. Areas with lagging urbanization accounted for the vast majority of Qinghai, accounting for more than 99.5%, and showed a downward trend from 1995 to 2020, with the largest decline from 2000 to 2005, reaching 0.04%. The proportion of areas with lagging ecosystem health showed an increasing trend year-on-year. The proportion of areas with lagging ecosystem health increased by 0.05% from 1995 to 2020, with the

largest increase in 2010–2015, reaching 47.76%. The proportion of areas with synchronous development showed an increasing trend. From 1995 to 2020, the proportion of areas with synchronous development of urbanization and ecosystem health increased by 0.04%, with the largest increase in 2000–2005, reaching 143.33% (Figure 6b). Areas with lagging ecosystem health are located in urban areas such as Xining, Delingha, and Geermu.

The spatial evolution of the *D* value between ecosystem health and urbanization in Qinghai during 1995–2020 is shown in Figure 7. The *D* value between urbanization and ecosystem health in Qinghai increased from 1995 to 2020. The area of seriously unbalanced development continued to decline from 1995 to 2010, increased slightly from 2010 to 2015, and then decreased significantly thereafter. The area of unbalanced development continued to increase from 1995 to 2010, showing a downward trend from 2010 to 2015, and increased significantly since then. In addition, the areas of slightly unbalanced development, moderately balanced development, and highly balanced development showed an upward trend during 1995–2020. Overall, the *D* value between ecosystem health and urbanization in Qinghai shows an increasing trend. Areas with a high *D* value are basically located in Xining, Delingha, Geermu, and other cities and their surrounding areas as well as areas with developed transportation infrastructure, showing a trend of expansion year-on-year. The areas with slightly unbalanced development are located at the periphery of the region with high *D* values, also showing a trend of expansion year-on-year. The low *D* value areas are located in the northwest and southwest of Qinghai, bounded by the Qinghai section of the Qinghai-Tibet Highway.

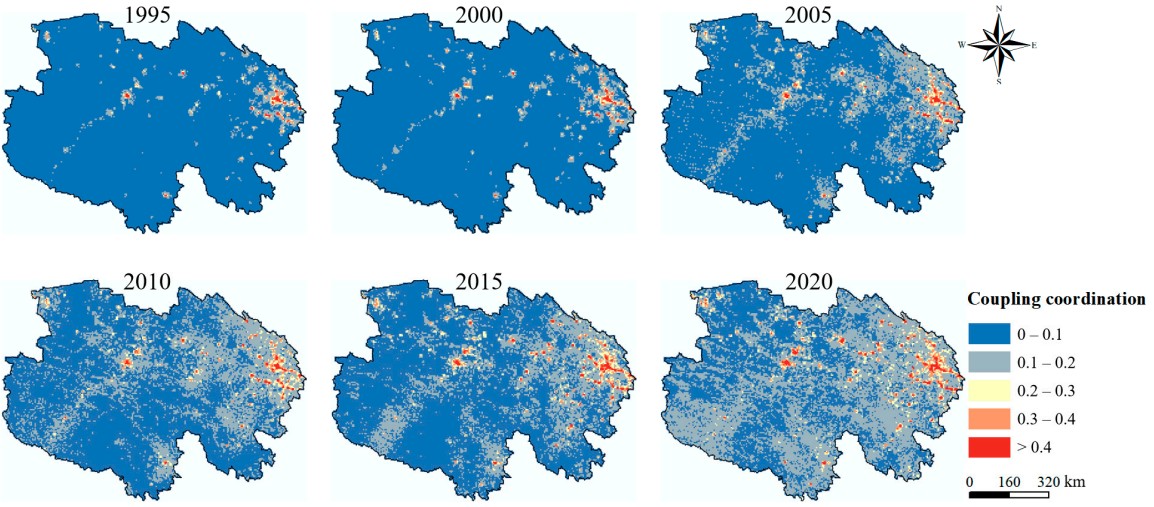

**Figure 7.** Spatial evolution of the *D* value between urbanization and ecosystem health.

## 4. Discussion

### 4.1. Spatiotemporal Characteristics of EHI

Identifying factors affecting ecosystem health is important for both scientific research and policy practice [2,23]. During 1995–2020, the ecosystem health pressure level in Qinghai continued to rise, the level of ecosystem health state continued to improve, and the fluctuation of EHI in Qinghai continued to increase, reaching its peak in 2020. The east and south of the study area are high-EHI-value areas, the northwest has low-EHI-value areas, and the rest are transitional areas. The natural environment in EHI low-value areas is relatively harsh, and the main land types are unused land, such as sandy land, Gobi, saline alkali land, swampy land, and exposed rocks. These types have low vegetation coverage and are difficult to utilize [57]. Regions with high EHI values have relatively suitable natural environments, low natural pressures, and relatively stable ecosystems [3]. The average patch size of areas with low levels of human interference, such as Qinghai Lake, Sanjiangyuan National Park, and the northern part of the study area, is relatively large, and landscape fragmentation is relatively low. In areas with intensive human activities

and high development levels such as cities and agricultural and pastoral areas, the average patch area of the landscape is small, the landscape structure is complex, and the degree of landscape fragmentation is high [58], such as the agricultural area in eastern Qinghai. During 1995–2020, the proportional difference between different landscape types decreased, the diversity of Qinghai's landscapes increased, and the degree of landscape fragmentation lessened. Figure 8a visualized the positive incremental changes in ecosystem health in Qinghai from 1995 to 2020, indicating that the regions with increased EHI are widely distributed throughout Qinghai. During 1995–2020, the conversion of other land use types in Qinghai to grassland and forest accounted for 57.80% of the total conversion area of LULC. He et al.'s research shows that the change of ecosystem health is closely related to ecological protection factors on the nearby Loess Plateau. This also proves that ecological protection efforts are effective for vegetation restoration and ecosystem health improvement [3]. The implementation of ecological protection projects such as the Grain-for-Green Program and the Three-North Shelterbelt Project greatly improved the NDVI of Qinghai, improved the vegetation status in the study area [38], and increased the EHI. In areas with frequent human activities, land was mainly transformed into farmland and build-up land.

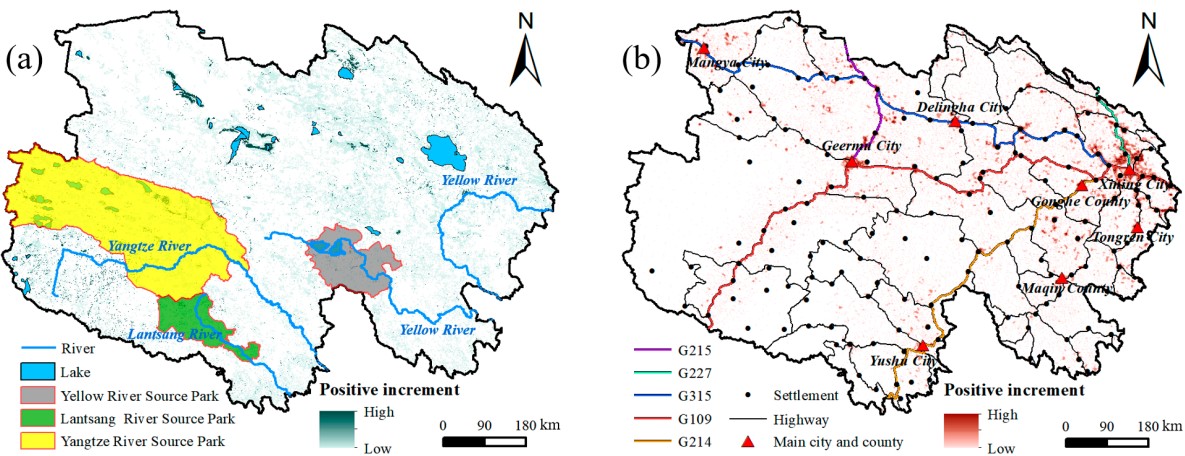

**Figure 8.** (**a**) Positive incremental changes in ecosystem health from 1995 to 2020; (**b**) positive incremental changes in urbanization from 1995 to 2020.

National parks are a type of protected area, a geographical space with clear boundaries that aims to protect the originality and integrity of the ecosystem, and have comprehensive functions such as scientific research, education, and leisure, providing an important way to systematically manage natural resources [59]. Establishing a nature reserve system with national parks as the mainstay is an important aspect of ecological civilization construction in China and an important measure to modernize environmental governance. The relationships between environment, economy, and society in the Sanjiangyuan area are complex and the tensions are increasingly prominent. The Sanjiangyuan Nature Reserve was established in 2000, which made a significant contribution to the ecological protection of Qinghai [60]. In 2016, Sanjiangyuan National Park, as China's first national park, launched a pilot project in Qinghai, which helped Qinghai further advance its ecological protection efforts [61]. In view of the importance of national parks, this study constructed a framework with Sanjiangyuan National Park as the evaluation unit while exploring the health status of the ecosystem in Qinghai. The National Park, located in the hinterland of the Tibetan Plateau and in the south of Qinghai Province has a total area of 123,100 km$^2$. The landform is dominated by mountains and valleys, with an average elevation of more than 4500 m. The park is rich in biodiversity and has great ecological value. As shown in Table 5, the EHI of Sanjiangyuan National Park is rising in fluctuations. Among them, the ecosystem health of the Yangtze River headwaters area is the best, followed by the Yellow River headwaters area and the Lantsang River headwaters area. However, the ecosystem health of all three

headwaters areas has been improving. Under the influence of China's ecological protection efforts, the EHI value in eastern Qinghai is gradually increasing, especially in Qinghai Lake, Zhaling Lake, Eling Lake, and other lakes and surrounding areas.

**Table 5.** Changes in the EHI of Sanjiangyuan National Park from 1995 to 2020.

| Region | Year | | | | | |
|---|---|---|---|---|---|---|
| | 1995 | 2000 | 2005 | 2010 | 2015 | 2020 |
| Yangtze River Source Park | 0.0558 | 0.0584 | 0.0616 | 0.0606 | 0.0668 | 0.0657 |
| Lantsang River Source Park | 0.0348 | 0.0348 | 0.0359 | 0.0343 | 0.0414 | 0.0457 |
| Yellow River Source Park | 0.0486 | 0.0459 | 0.0476 | 0.0476 | 0.0674 | 0.0629 |
| Total | 0.1392 | 0.1391 | 0.1451 | 0.1425 | 0.1756 | 0.1743 |

*4.2. Urbanization Characteristics*

From 1995 to 2000, the level of urbanization in Qinghai gradually increased, and the urbanization index grew by 16.54%. The urbanization index rose the most from 2000 to 2005, reaching 51.11%. Since then, the increase of the urbanization index gradually slowed down, with the smallest rise of 1.16% between 2005 and 2010. The urbanization index increased slightly from 2010 to 2015, reaching 20.72%. The urbanization index rose by 14.95% from 2015 to 2020.

Since 2000, the urbanization process in Qinghai has accelerated. Since the implementation of the country's Western Development Strategy in 2000, the development of major projects such as salt lakes and oil and gas resources in Qinghai has led to an increase in population and materials flows. Since then, the increase of the urbanization index has gradually slowed. Overall urbanization level in Qinghai is not high, with a large proportion of low- and lower-level urbanization areas. Areas with a high level of urbanization radiate outward from major cities such as Xining, Geermu, and Delingha. The urbanization of the Qaidam region in western Qinghai laid the foundation for the urbanization of Geermu and Delingha. In the 1960s, during the "Second Five Year Plan" period, a large number of highly industrialized projects were relocated to Xining and its surrounding areas, resulting in an increase in the level of urbanization [62]. From 1995 to 2020, the area of low- and lower-level urbanization areas in Qinghai has been decreasing, the area of medium-level urbanization areas has increased significantly, and the area of high- and higher-level urbanization has also increased significantly. Figure 8b visualizes the positive incremental changes in urbanization in Qinghai from 1995 to 2020 and shows the distribution of major cities, counties, settlements, and highways in the study area. It can be seen that the dynamic changes in Qinghai's urbanization level are influenced by transportation and terrain, and the urbanization level is negatively correlated with altitude. Areas with higher urbanization levels basically overlap with the distribution of settlements. The construction status of highways also strongly determines the distribution trend of urbanization [38], especially around National Highway 109, which clearly shows the influence of transport infrastructure on urbanization in this relatively remote and sparsely populated region.

*4.3. Spatiotemporal Heterogeneity of Coupling between Ecosystem Health and Urbanization*

During the study period, except for areas with severely unbalanced development, which showed a downward trend in fluctuations, other areas basically showed an upward trend, namely, the *D* value between ecosystem health and urbanization in Qinghai continued to strengthen, gradually shifting towards a balanced development direction. Qinghai is located at a high elevation, far from coastal ports, and far from developed regions, making it difficult to form effective networks of social and economic exchange. Therefore, the process of urbanization in Qinghai faces the location problem of insufficient development. However, the national strategy of developing China's western region promoted the urbanization development of Qinghai. From 1995 to 2020, the area of urbanization lagging regions continued to decrease, with the largest decline from 2000 to 2005. The growth of urbanization has increased the pressure on the eco-environment, and many areas have

begun to lag in ecosystem health. The proportion of areas with lagging ecosystem health is increasing year-on-year. The proportion of areas with coordinated development also shows a trend of increasing year-on-year. Areas with lagging ecosystem health are located in urban areas such as Xining, Delingha, and Geermu. The coordinated development areas are located around the ecosystem health lagging area. The eco-environment in these areas is well protected and can withstand the higher development of urbanization. High *D* value areas are basically located in major cities and their surrounding areas, as well as areas with developed transportation, and show an increasing trend year-on-year. Areas with slightly unbalanced development are located on the periphery of areas with high *D* value and also show a trend of expanding year-on-year. The low *D* value areas are located in the northwest and southwest of Qinghai, bounded by the Qinghai section of the Qinghai-Tibet Highway.

*4.4. Limitations and Future Research*

Qinghai is an important area for the Belt and Road Initiative [34] and a core area of the Tibetan Plateau [8]. Therefore, exploring the coupling coordination relationship between urbanization and ecosystem health in Qinghai has important practical significance. We visualize the regional differences of ecosystem health and urbanization in Qinghai and their key driving factors. The innovation of this study is to visually analyze the relationship between ecosystem health and urbanization in Qinghai from a grid scale. This study shows that during the period 1995–2020, the *D* value between urbanization and ecosystem health in Qinghai continued to improve. This study can provide ideas for research on the coupling coordination of urbanization and ecological health in other high-altitude regions. When undertaking ecological protection measures in such areas, we should fully consider the natural driving factors and socio-economic driving factors and formulate ecological protection policies in line with local environmental conditions. For example, meteorological disaster monitoring can prevent and reduce the impact of weather conditions on ecosystems, change vegetation types to improve vegetation coverage, and monitor and evaluate the ecosystem services generated by restoration projects [63,64]. However, this study also faced uncertainties and limitations. In this study, the main factors affecting ecological health in Qinghai were selected, but in the future, more detailed data and the weight setting of each factor could be improved.

## 5. Conclusions

This study analyzes the evolution of land use in Qinghai based on the land use transfer matrix. The research shows that the land use types in Qinghai are mainly grassland and unused land, accounting for about 90% of the total area of Qinghai. During 1995–2020, land use in Qinghai underwent major shifts, including significant changes in grassland and unused land; the conversion of other land use types to grassland and unused land occurred across the province.

The changes in stress and state indicators of EHI indicate that during 1995–2020, although the stress level in Qinghai was increasing, the state level improved. The level of ecosystem health has shown a fluctuating upward trend, with the EHI value in 2020 increasing by 3.31% compared to 1995. The overall distribution trend of EHI is that the east and south of Qinghai are high-value areas, the northwest has low-value areas, and the rest are transitional areas. Over time, the EHI value in the eastern part of Qinghai gradually increased, especially in Qinghai Lake, Zhaling Lake, Eling Lake, and other lakes and surrounding areas. The level of urbanization in Qinghai continues to rise, with the urbanization index in 2020 increasing by 147.20% compared to 1995. The urbanization level in Qinghai is generally not high. The proportion of low- and lower-level urbanization areas exceeded 95%. Around Xining in the southeast and Geermu in the west of Qinghai, high-level areas are expanding radially.

The CCD between urbanization and ecosystem health in Qinghai is on the rise. Areas with high *D* value are mainly located in major urban agglomerations and their surrounding areas, as well as areas with developed transportation, and show an increasing trend year-

on-year. Areas with slightly unbalanced development are located on the periphery of high *D* value regions and also show a trend of expanding year-on-year. The low *D* value areas are located in the northwest and southwest of Qinghai, bounded by the Qinghai section of the Qinghai-Tibet Highway. Areas with lagging urbanization account for the vast majority of Qinghai. From 1995 to 2020, the area of urbanization lagging areas showed a downward trend, while the proportion of ecosystem health lagging areas showed an upward trend year-on-year. The proportion of areas with lagging ecosystem health increased by 0.05%, while the proportion of areas with coordinated development showed an upward trend year-on-year. The proportion of areas with simultaneous urbanization and ecosystem health development increased by 0.04%. Areas with lagging ecosystem health are located in urban areas such as Xining, Delingha, and Geermu. The coordinated development areas are located around the ecosystem health lagging area, while other areas in Qinghai are urbanization lagging areas.

**Author Contributions:** Conceptualization, Y.H. and L.G.; methodology, Y.H., L.G. and S.Z.; software, Y.H.; validation, Y.H. and L.G.; formal analysis, Y.H.; investigation, Y.H., T.W. and L.G.; resources, T.W. and L.G.; data curation, Y.H. and S.Z.; writing—original draft preparation, Y.H.; writing—review and editing, Y.H., T.W., L.G. and S.Z.; visualization, Y.H.; supervision, L.G.; project administration, L.G.; funding acquisition, L.G. All authors have read and agreed to the published version of the manuscript.

**Funding:** This research was funded by the National Natural Science Foundation of China, grant number No. 32271666 and the Second Tibetan Plateau Scientific Expedition and Research (STEP) program, grant number 2019QZKK0308.

**Institutional Review Board Statement:** Not applicable.

**Informed Consent Statement:** Not applicable.

**Data Availability Statement:** The data that support the findings of this study are available from the corresponding author upon reasonable request.

**Conflicts of Interest:** The authors declare no conflict of interest.

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
