# Peer review of "Spatiotemporal Relationships between Ecosystem Health and Urbanization on the Tibetan Plateau from a Coupling Coordination Perspective"

_land, doi:10.3390/land12081635_

Round 1

Reviewer 1 Report

The article deals with a complex issue. The authors approached the task by addressing the relationship between urbanization and ecosystem health through the assessment based on a series of well-known and commonly used indexes. They referred to instances of these indexes being used in other studies. However, it still does not provide a simple and straightforward answer as to whether this is the path to achieving a sufficiently comprehensive response. Nevertheless, I appreciate that they took on such a challenging task.

The article is structured in a clear and organized manner, with well-established chapters. The chosen time frames and intervals are appropriate and satisfactory. The data used in the study are from relevant sources. The cited literature mostly refers to authors from Asia, while the used indexes are a common part of the Western academic environment from which they originated.

What puzzles me is the assertion that the conversion of land to higher urbanization has a positive effect on the landscape. Landscape fragmentation may indeed lead to increased diversity, as long as it does not involve artificial elements. Were areas with increasing urbanization or artificially planted vegetation assessed as areas that enhance biodiversity? While under certain circumstances, this might be the case, it seems more likely to result in the opposite trend. When working with data, one should differentiate between the ecological value of built-up areas and rural environments. I am not familiar with the situation in this part of China, and it may be unique, but often, fragmented smaller agricultural areas with traditional farming practices provide greater eco-stability compared to consolidated areas.

The indexes included in the final evaluation are relevant, but they could be influenced by human interventions in the system, such as irrigation, expansion of artificially planted vegetation, which lacks stability comparable to natural vegetation without regular interventions, natural selection, and so on. I point this out because, although the EHI (Ecosystem Health Index) shows a positive trend, it might be a distorted result. In my opinion, it would be beneficial to include an indicator related to the extent of preserved natural vegetation to make the evaluation more objective. However, I cannot assess the extent to which such data is available in the evaluated province.

Round 2

Reviewer 1 Report

The authors' argumentation is convincing, and I believe the article is ready for publication in its current form. For the assessment of such an extensive territory, it is indeed necessary to choose methods that do not focus on the large-scale details.